# Shape-Factor Impact on a Mass-Based Hybrid Nanofluid Model for Homann Stagnation-Point Flow in Porous Media

**DOI:** 10.3390/nano13060984

**Published:** 2023-03-08

**Authors:** Shiyuan Li, Xiangcheng You

**Affiliations:** 1College of Petroleum Engineering, China University of Petroleum-Beijing, Beijing 102249, China; 2State Key Labortory of Petroleum Resources and Prospecting, Beijing 102249, China

**Keywords:** mass-based hybrid nanofluids, Homann stagnation-point flows, shape factor, porous media

## Abstract

This paper studies the impact of shape factor on a mass-based hybrid nanofluid model for Homann stagnation-point flow in porous media. The HAM-based Mathematica package BVPh 2.0 is suitable for determining approximate solutions of coupled nonlinear ordinary differential equations with boundary conditions. This analysis involves discussions of the impact of the many physical parameters generated in the proposed model. The results show that skin friction coefficients of *Cf_x_* and *Cf_y_* increase with the mass of the first and second nanoparticles of the hybrid nanofluids *w*_1_ and *w*_2_ and with the coefficient of permeability in porous media. For the axisymmetric case of γ = 0, when *w*_1_ = *w*_2_ = 10 gr, *w*_*f*_ = 100 gr and *Cf_x_* = *Cf_y_* = 2.03443, 2.27994, 2.50681, and 3.10222 for *σ* = 0, 1, 2, and 5. Compared with *w*_1_ = *w*_2_ = 10 gr, *w*_f_ = 100 gr, and *σ* = 0, it can be found that the wall shear stress values increase by 12.06%, 23.21%, and 52.48%, respectively. As the mass of the first and second nanoparticles of the mass-based hybrid nanofluid model increases, the local Nusselt number Nu_x_ increases. Values of Nu_x_ obviously decrease and change with an increase in the coefficient of permeability in the range of γ < 0; otherwise, Nu_x_ is less affected in the range of γ > 0. According to the calculation results, the platelet-shaped nanoparticles in the mass-based hybrid nanofluid model can achieve maximum heat transfer rates and minimum surface friction.

## 1. Introduction

A hybrid nanofluid consists of two kinds of nanofluids combined in a base fluid. Hybrid nanofluids have become a hot field of interest for researchers in engineering applications of various disciplines and industries, such as the heat transfer of nanofluids in micro-channel or porous media, geothermal applications, oil-flow filtration, and so on [1,2,3,4,5,6,7,8].

It is of great interest to researchers to apply hybrid nanoparticles and to determine how the shape of nanoparticles influences thermophysical properties. Murshed et al. [9] used deionized water as the medium to prepare nanofluids from spherical and rod-shaped TiO_2_ nanoparticles. In addition to nanoparticle volume fraction, nanoparticle sizes and shapes also contribute to the enhancement of thermal conductivity. Alumina nanoparticles of different shapes were examined by Timofeeva et al. [10] in a mixture of ethylene glycol and water to determine their thermal conductivity and viscosity. The viscosity and thermal conductivity of nanofluids with approximate rectangular and spherical shapes were studied experimentally by Jeong et al. [11]. The significant effects of nanoparticle shapes on viscosity and thermal conductivity were observed in the volume concentration range of 0.05–5.0 vol%. In a study by Elias et al. [12], different nanoparticle shapes were examined with regard to the performance of shell-and-tube heat exchangers using five shapes of nanoparticles. As a result, the cylindrical nanoparticles demonstrated better heat transfer characteristics and an even higher rate of heat transfer. An analysis of heat transfer and fluid-flow characteristics was conducted by Vanaki et al. [13] using SiO_2_ nanoparticles of different concentrations and shapes. In terms of heat transfer enhancement, SiO_2_-EG nanofluids with platelet-shaped nanoparticles exhibited the highest performance. Using a stretch sheet, Ghadikolaei et al. [14] studied the effects of induced magnetic fields on stagnant flows of hybrid nanofluids. The effect of temperature distribution on the shape of hybrid nanofluids was studied. Generally, platelet-shaped nanoparticles proved to be more effective than brick-shaped, cylindrical, and spherical nanoparticles. The effects of magnetic force and radiation on alumina migration in permeable media were simulated by Sheikholeslami et al. [15] using a new numerical method. In order to enhance the characteristics of the working liquid, Al_2_O_3_-water with various nanoparticle shapes was selected. In an isothermal heated horizontal tube, Benkhedda et al. [16] conducted numerical simulations of the steady-state forced-convection heat transfer and fluid-flow characteristics of hybrid nanofluids of various shapes. Nanoparticles with bladed shapes exhibited the highest heat transfer rate when the volume concentration was high, followed by those with platelet, cylindrical, or spherical shapes. Depending on the different shapes and radiation levels of the nanoparticles, the hydrothermal properties of Al_2_O_3_-H_2_O nanofluids passing through a porous shell under ambient magnetic conditions were studied by Shah et al. [17] and Khashi’ie et al. [18]. You et al. [19,20,21] studied the flow characteristics of Cu-Al_2_O_3_-H_2_O hybrid nanofluids in the inclined microchannel of porous media. A smaller average spherical particle size and a high concentration of small particles enhanced the heat transfer within the nanofluid. Wanatasanappan et al. [22] carried out experimental research on the viscosity and rheological properties of hybrid nanofluids, analyzed the influence of Al_2_O_3_-Fe_2_O_3_ mixing ratios on viscosity properties, and established a correlation with viscosity prediction. Using a rotating disk with a constant radial stretching rate, Dinavand et al. [23] explored the three-dimensional laminar flow of a hybrid nanofluid in an incompressible, steady condition. Calculations for the hybrid single-phase nanofluid model were based on the mass of nanoparticles in conjunction with the mass of the base fluid at a constant pressure. For the flow of an incompressible, two-dimensional, hybrid nanofluid on a convection-heated moving wedge with a radiation transition, Berrehal et al. [24] calculated the steady flow using numerical simulation. Spherical and non-spherical nanoparticle suspensions of magnetite (Fe_3_O_4_) and graphene oxide (GO) were suspended in pure water. Rahimi et al. [25] studied two-dimensional natural convection and entropy generation in a hollow heat exchanger filled with a CuO-water nanofluid. The KKL model was used to estimate the dynamic viscosity of nanoparticles based on their shape in the simulation. In nanofluid-filled channels, Rao et al. [26] considered fluid flow, heat transfer, entropy generation, and hot-wire visualization using the finite volume method. A Koo–Kleinstreuer–Li model was used to estimate dynamic viscosity, and Brownian motion was taken into account. In horizontal microchannels, Soumya et al. [27] examined the flow and thermal properties of Fe_3_O_4_-Ag/water and Fe_3_O_4_-Ag/kerosene hybrid nanofluids and analyzed the effect of different nanoparticle shape factors on nanofluid temperature. The perturbation technique was used by Subray et al. [28] to study the effect of the nanoparticle shape factor on convective heat and mass transfer in an inclined pipe. The thermal conductivity of SWCNT-CuO (25:75)/water nanofluids was investigated by Esfe et al. [29] using basic parameters such as temperature and the solid volume fraction. A permeable exponentially shrinking Riga surface with thermal radiation energy was considered by Mandal et al. [30] to determine the flow of hybrid Ag-MoS_2/water nanofluids. They investigated the velocity, temperature, surface friction coefficients, the Nusselt number, and entropy generation at the contraction Riga surface under convective heat boundary conditions, as well as the way hybrid nanofluids varied in viscosity, thermal conductivity, and slip velocity. Farooq et al. [31] studied the velocity, thermal field, and entropy distribution characteristics of hybrid nanofluids when passing through a thermal radiation slurry. With Cattaneo–Christov heat flux, carbon nanotubes were used as nanoparticles, and ethylene glycol was used as a base fluid. Utilizing X-ray-computed tomography and 3D scanning transmission electron microscopy, Li et al. [32] characterized the combined effects of nanofiller volume fractions and packer–polymer interface interactions. As a means of improving heat transfer capacity, Qi et al. [33] developed a contact probability model for analyzing silicone rubber composites with hybrid fillers in terms of thermal conductivity. In both experimental and simulation studies, the volume fraction, filler shape, and filler size were found to be the most significant factors that affect a composite material’s thermal conductivity. Using water-based Fe_3_O_4_-Al_2_O_3_-ZnO nanofluid, Adun et al. [34,35,36] investigated the effect of temperature, volume concentration, and mixing ratio on the fluid. Machine learning models were also developed for predicting the fluid’s characteristics. Similarly, ternary hybrid nanofluids were studied for thermal conductivity and dynamic viscosity.

There are many references on stagnation point flow, some of which are studied in the following literature survey. Ariel [37] studied the two-dimensional stagnation-point flow problem of second-order non-Newtonian fluids. Weidman [38,39] changed exterior potential flow in Homann’s problem and solved non-axisymmetric stagnation-point flows and rotational stagnation-point flows. Dinarvan et al. [40] solved Tiwari–Das nanofluid models by using Homotopy Analysis Method (HAM), and observed transient MHD stagnation-point and heat transfer over a vertically permeable sheet for nanofluids. Othman et al. [41] investigated numerically steady flow of two-dimensional mixed convection boundary layers near stagnation on impermeable vertical surfaces that are stretching and shrinking. It was investigated by Abbas et al. [42] whether stagnation-point flows occurred in MHD micropolar nanomaterial fluid flowing around a sinusoidally shaped cylinder. In addition, the velocity slip of porous surfaces was also studied. Turkyilmazoglu [43] mainly used numerical and perturbation methods to study the unsteady flow field caused by the deceleration of a rotating ball. A vertically stretched thin plate was examined by Sharma et al. [44] for effects of heat generation and absorption on mixed-convection stagnation-point flows with external magnetic fields. The magneto-hydrodynamic oscillatory oblique stagnation-point flows of micropolar nanofluids were analyzed by Sadiq et al. [45]. Copper and alumina nanoparticles were studied while the aqueous base solution was observed. Ahmed et al. [46] used Tiwari and Das models to study heat transfer characteristics of hybrid nanofluids in non-axisymmetric Homann stagnation region with magnetic flux. The importance of the shape factors of nanoparticles, namely cylinders, blades, bricks and platelets, was studied under free flow conditions independent of time. Khan et al. [47] considered unsteady three-dimensional non-axisymmetric Homann flows of conducting nanofluids under buoyancy. By using the fourth-order Runge–Kutta method combined with shooting techniques, Mahapatra et al. [48] developed a numerical method to solve nonaxisymmetric Homann stagnation-point flows on a rigid plate of viscoelastic fluid. Khan et al. [49] studied Homann stagnation-point flows of non-axisymmetric Walter’s B nanofluids, and cylindrical disk exhibited nonlinear Rosseland thermal radiation and magnetohydrodynamics that was independent of time. As described in Waini et al. [50], hybrid nanofluid flows on a flat plate with non-axisymmetric stagnation points are studied.

This study applies the homotopy analysis method (HAM) to approximate analytical solutions for shape-factor impact on a mass-based hybrid nanofluid model for Homann stagnation-point flow in porous media. Some HAM-based packages developed in Maple or Mathematica simplify its application. The free software BVPh 2.0 can be downloaded online (http://numericaltank.sjtu.edu.cn/BVPh.htm (accessed on 18 May 2013)). It is an easy-to-use tool that calculates boundary layer flows [51,52]. This paper is divided into four sections. In addition to the introduction in Section 1, Section 2 contains mathematical descriptions of this problem. Section 3 covers the results and discussion and includes graphic illustrations and tables. Finally, Section 4 contains the conclusions, highlighting the main findings in this work.

## 2. Mathematical Formulas

A mass-based hybrid nanofluid model for Homann stagnation-point flow in porous media is shown in Figure 1. The $\bar{z}$-axis represents normal direction and $\bar{x}\bar{y}$ represents a plane in Cartesian coordinate systems. The governing equations are as follows (Weidman [38]; Waini et al. [50]):(1)$$\frac{\partial \bar{u}}{\partial \bar{x}}+\frac{\partial \bar{v}}{\partial \bar{y}}+\frac{\partial \bar{w}}{\partial \bar{z}}=0,$$
(2)$$\bar{u}\frac{\partial \bar{u}}{\partial \bar{x}}+\bar{v}\frac{\partial \bar{u}}{\partial \bar{y}}+\bar{w}\frac{\partial \bar{u}}{\partial \bar{z}}={\bar{u}}_{e}\frac{d{\bar{u}}_{e}}{d\bar{x}}+\frac{\mu_{hnf}}{\rho_{hnf}}\frac{\partial^{2}\bar{u}}{\partial {\bar{z}}^{2}}-\frac{\mu_{hnf}}{K\rho_{hnf}}\left(\bar{u}-{\bar{u}}_{e}\right),$$
(3)$$\bar{u}\frac{\partial \bar{v}}{\partial \bar{x}}+\bar{v}\frac{\partial \bar{v}}{\partial \bar{y}}+\bar{w}\frac{\partial \bar{v}}{\partial \bar{z}}={\bar{v}}_{e}\frac{d{\bar{v}}_{e}}{d\bar{x}}+\frac{\mu_{hnf}}{\rho_{hnf}}\frac{\partial^{2}\bar{v}}{\partial {\bar{z}}^{2}}-\frac{\mu_{hnf}}{\bar{K}\rho_{hnf}}\left(\bar{v}-{\bar{v}}_{e}\right),$$
(4)$$\bar{u}\frac{\partial \bar{T}}{\partial \bar{x}}+\bar{v}\frac{\partial \bar{T}}{\partial \bar{y}}+\bar{w}\frac{\partial \bar{T}}{\partial \bar{z}}=\frac{{\bar{k}}_{hnf}}{{\left(\rho c_{p}\right)}_{hnf}}\frac{\partial^{2}\bar{T}}{\partial {\bar{z}}^{2}},$$
subject to
(5)$$\begin{array}{l} \bar{u}=0,\;\bar{v}=0,\;\bar{w}=0,\;\bar{T}=\begin{array}{cc} {\bar{T}}_{w}\left(x\right) & \begin{array}{cc} at & \bar{z}= \end{array} \end{array}0\\ \bar{u}\to {\bar{u}}_{e}\left(x\right),\;\bar{v}\to {\bar{v}}_{e}\left(x\right),\;\bar{w}\to {\bar{w}}_{e}\left(x\right),\;\bar{T}\to \begin{array}{cc} {\bar{T}}_{\infty } & \begin{array}{cc} at & \bar{z}\to \end{array} \end{array}\infty \end{array},$$

In the formulas, $\bar{u},\bar{v},\bar{w}$ represent velocity components, external flow velocities are ${\bar{u}}_{e}\left(\bar{x},\bar{y}\right)=\bar{x}\left(\bar{a}+\bar{b}\right)$, ${\bar{v}}_{e}\left(\bar{x},\bar{y}\right)=\bar{y}\left(\bar{a}-\bar{b}\right)$, and ${\bar{w}}_{e}\left(\bar{z}\right)=-2\bar{a}\bar{z}$ such that $\bar{a},\bar{b}$ represent shear–strain rates; surface temperature is represented by ${\bar{T}}_{w}={\bar{T}}_{\infty }+{\bar{T}}_{0}\bar{x}$ (${\bar{T}}_{0}$ represents characteristic temperature and ${\bar{T}}_{\infty }$ represents ambient temperature); $\bar{K}$ represents the permeability of porous media; $\bar{k}$ represents conductivity at certain temperatures; ρ represents fluid density; $c_{p}$ represents the specific heat capacity coefficient; and μ represents dynamic viscosity. Table 1 shows the thermophysical properties of H_2_O, Cu, and Al_2_O_3_ nanoparticles. Table 2 and Table 3 present mass-based hybrid nanofluid models for the thermophysical properties of spherical nanoparticles.

It is worth noting that $w_{1},w_{2},w_{f}$ are the masses of the first and second nanoparticles and base fluid water, respectively; $\phi_{1}$,$\phi_{2}$ represent Cu and Al_2_O_3_ nanoparticles, respectively; and solid compositions are represented by the subscripts n1 and n2. $\phi =\phi_{1}+\phi_{2}$ is the volume fraction of hybrid nanofluids, and n is the shape factor, n=3/ψ, where ψ represents the sphericity of nanoparticles. Table 4 shows A and B coefficients for non-spherical nanoparticles in the effective viscosity relation for nanoparticles of different shapes.

Using similarity transformation (Weidman [38]; Waini et al. [50]), we obtain
(6)$$\begin{array}{l} \bar{u}=\bar{x}f\prime \left(\eta \right)\left(\bar{a}+\bar{b}\right),\;\bar{v}=\bar{y}g\prime \left(\eta \right)\left(\bar{a}-\bar{b}\right),\;\bar{w}=-\sqrt{\bar{a}\nu_{f}}\left[f\left(\eta \right)\left(\bar{a}+\bar{b}\right)+g\left(\eta \right)\left(\bar{a}-\bar{b}\right)\right]\\ \theta =\frac{\bar{T}-{\bar{T}}_{\infty }}{{\bar{T}}_{w}-{\bar{T}}_{\infty }},\;\eta =\bar{z}\sqrt{\frac{\bar{a}}{\nu_{f}}} \end{array},$$

In the formula, ’ indicates the derivative with respect to η.

Substitute Equation (6) into Equations (1)–(4), and one gets:(7)$$N_{f}\left(f,g\right)=A_{1}f\prime \prime \prime +\left(1+\gamma \right)\left(ff\prime \prime +1-f\prime^{2}\right)+\left(1-\gamma \right)gf\prime \prime -A_{1}\sigma \left(f\prime -1\right)=0,$$
(8)$$N_{g}\left(f,g\right)=A_{1}g\prime \prime \prime +\left(1-\gamma \right)\left(gg\prime \prime +1-g\prime^{2}\right)+\left(1+\gamma \right)fg\prime \prime -A_{1}\sigma \left(g\prime -1\right)=0,$$
(9)$$N_{\theta }\left(f,g,\theta \right)=\frac{A_{2}}{\mathrm{Pr}}\theta \prime \prime +\left(1+\gamma \right)\left(f\theta \prime -f\prime \theta \right)+\left(1-\gamma \right)g\theta \prime =0,$$
subject to
(10)$$\begin{array}{l} f\left(0\right)=0,\;f\prime \left(0\right)=0,\;g\left(0\right)=0,\;g\prime \left(0\right)=0,\;\theta \left(0\right)=1\\ f\prime \left(\eta \right)\to 1,\;g\prime \left(\eta \right)\to 1,\;\theta \left(\eta \right)\to \begin{array}{cc} 0 & \begin{array}{cc} at & \eta \to \end{array} \end{array}\infty \end{array},$$

In the formula, $N_{f}\left(f,g\right),\;N_{g}\left(f,g\right),\;N_{\theta }\left(f,g,\theta \right)$ are nonlinear differential operators; the reflective symmetries are obtained through f(η,γ)=g(η,−γ) or f(η,−γ)=g(η,γ), where γ is the ratio of the strain–shear rate. The coefficients of $A_{1},\;A_{2}$, σ, and Pr are given by
(11)$$A_{1}=\frac{\mu_{hnf}/\mu_{f}}{\rho_{hnf}/\rho_{f}},\;A_{2}=\frac{k_{hnf}/k_{f}}{{\left(\rho c_{p}\right)}_{hnf}/{\left(\rho c_{p}\right)}_{f}},\;\gamma =\frac{b}{a},\;\sigma =\frac{\nu_{f}}{aK},\;\mathrm{Pr}=\frac{\mu_{f}{\left(c_{p}\right)}_{f}}{k_{f}}.$$

The skin friction coefficients of $Cf_{x},\;Cf_{y}$ and the local Nusselt number $Nu_{x}$ are
(12)$${\bar{C}}_{fx}=\frac{\mu_{hnf}}{\rho_{f}u_{e}^{2}}{\left(\frac{\partial \bar{u}}{\partial \bar{z}}\right)}_{\bar{z}=0},\;{\bar{C}}_{fy}=\frac{\mu_{hnf}}{\rho_{f}v_{e}^{2}}{\left(\frac{\partial \bar{v}}{\partial \bar{z}}\right)}_{\bar{z}=0},\;{\bar{Nu}}_{x}=-\frac{\bar{x}k_{hnf}}{k_{f}\left({\bar{T}}_{w}-{\bar{T}}_{\infty }\right)}{\left(\frac{\partial \bar{T}}{\partial \bar{z}}\right)}_{\bar{z}=0},$$
(13)Cfx=Rex(1+γ)C¯fx=μhnfμff″(0), Cfy=Rey(1−γ)C¯fy=μhnfμfg″(0)Nux=1+γRexNu¯x=−khnfkfθ″(0),

In the formulas, the local Reynolds numbers are Rex=uex/νf, and Rey=vey/νf. HAM-based Mathematica packages [48,49] are suitable for determining approximate solutions of coupled nonlinear ordinary differential Equations (7)–(9) with Boundary Conditions (10). Free online instructions for BVPh 2.0 are available online (http://numericaltank.sjtu.edu.cn/BVPh.htm (accessed on 18 May 2013) ).

As a result, HAM based on topological homotopy transforms a nonlinear problem into an infinite linear subproblem without requiring any physical parameters. The problems considered have the following characteristics:(14)$$f(\eta )=\sum_{m=0}^{+\infty }f_{m}\left(\eta \right),\;g(\eta )=\sum_{m=0}^{+\infty }g_{m}\left(\eta \right),\;\theta (\eta )=\sum_{t=0}^{+\infty }\theta_{m}(\eta ),$$

In the formula, $f_{m}\left(\eta \right),\;g_{m}\left(\eta \right),\;\theta_{m}\left(\eta \right)$ are calculated based on the higher-order deformation equation controlled by the selected auxiliary linear operator. In accordance with Equations (7)–(9) and the Boundary Conditions (10) at infinity, f(η), g(η), θ(η) should be in the form
(15)$$f(\eta )=\sum_{k=0}^{+\infty }\sum_{i=0}^{+\infty }\sum_{j=0}^{+\infty }a_{i,j}^{k}\eta^{i}e^{-j\eta },\;g(\eta )=\sum_{k=0}^{+\infty }\sum_{i=0}^{+\infty }\sum_{j=0}^{+\infty }b_{i,j}^{k}\eta^{i}e^{-j\eta },\;\theta (\eta )=\sum_{k=0}^{+\infty }\sum_{i=0}^{+\infty }\sum_{j=0}^{+\infty }c_{i,j}^{k}\eta^{i}e^{-j\eta },$$
where $a_{i,j}^{k},\;b_{i,j}^{k},\;c_{i,j}^{k}$ are the constant coefficients to be determined by HAM-based Mathematica package BVPh 2.0. HAM relies heavily on the solution expression of Equation (15) to select auxiliary linear operators and initial guesses. It is important to note that the f(η), g(η), θ(η) given by BVPh 2.0 contain three unknown convergence control parameters, $d_{0}^{f},\;d_{0}^{g},\;d_{0}^{\theta }$. The series solution relies on these to ensure convergence. The mean residual errors of the kth-order approximations are defined as follows:(16)$$\zeta_{k}^{f}\left(d_{0}^{f},d_{0}^{g},d_{0}^{\theta }\right)=\frac{1}{N+1}{\sum_{i=0}^{N}\left[{N_{f}\left(\sum_{m=0}^{k}f_{m}\right)|}_{\eta =i\beta \eta }\right]}^{2},$$
(17)$$\zeta_{k}^{f}\left(d_{0}^{f},d_{0}^{g},d_{0}^{\theta }\right)=\frac{1}{N+1}{\sum_{i=0}^{N}\left[{N_{g}\left(\sum_{m=0}^{k}f_{m},\sum_{m=0}^{k}g_{m}\right)|}_{\eta =i\beta \eta }\right]}^{2},$$
(18)$$\zeta_{k}^{f}\left(d_{0}^{f},d_{0}^{g},d_{0}^{\theta }\right)=\frac{1}{N+1}{\sum_{i=0}^{N}\left[{N_{\theta }\left(\sum_{m=0}^{k}f_{m},\sum_{m=0}^{k}g_{m},\sum_{m=0}^{k}\theta_{m}\right)|}_{\eta =i\beta \eta }\right]}^{2},$$
for the original governing Equations (7)–(9). An approximation of the kth order has a total error defined as follows:(19)$$\zeta_{k}^{tol}\left(d_{0}^{f},d_{0}^{g},d_{0}^{\theta }\right)=\zeta_{k}^{f}\left(d_{0}^{f},d_{0}^{g},d_{0}^{\theta }\right)+\zeta_{k}^{g}\left(d_{0}^{f},d_{0}^{g},d_{0}^{\theta }\right)+\zeta_{k}^{\theta }\left(d_{0}^{f},d_{0}^{g},d_{0}^{\theta }\right),$$

As a result of the kth-order approximation, the optimal values for $d_{0}^{f},\;d_{0}^{g},\;d_{0}^{\theta }$ can be determined as the minimum of the total error of $\zeta_{k}^{tol}$. Consult the online BVPh 2.0 for details on specific operations (http://numericaltank.sjtu.edu.cn/BVPh.htm (accessed on 18 May 2013)).

## 3. Results Analysis and Discussion

This analysis involves a discussion of the impact of the many physical parameters generated in the proposed model. When γ=0, ϕ=0, and σ=0, it can be found that f(η)=g(η) represents the axisymmetric Homann stagnation-point flow. When γ=0 (axisymmetric), ϕ=0 (pure fluid), and σ=0 (non-porous medium), f″(0) = 1.311608, compared with f″(0) = 1.311938 in Waini et al. [38], the relative error is not more than 0.8194%. As shown in Table 5, the values of the skin friction coefficients of $Cf_{x},\;Cf_{y}$, and the local Nusselt number $Nu_{x}$ with γ, ϕ, and σ when Pr=6.2 are calculated compared with the results of Waini et al. [38]. Consequently, Table 5 demonstrates that the results of the present mass-based study ($w_{1}=w_{2}=0$, pure water) are consistent with previous similar work on volume fraction ($\phi =\phi_{1}=\phi_{2}=0$, pure water). All calculation cases are compared when the first (Cu) and second (Al_2_O_3_) nanoparticles have the same shape factor, namely $n_{1}=n_{2}$.

An analysis of the shape factor of nanoparticles as it relates to velocity distribution is shown in Figure 2, where $w_{1}=w_{2}=10\;gr\;,\;w_{f}=100\;gr$ = 0, σ = 0, Pr = 6.2, and γ = 0. First, velocity increases as the shape factor increases, and then it decreases as the shape factor decreases. Furthermore, disk-shaped nanoparticles ($n_{1}=n_{2}=8.3$) have similar velocity profiles to brick-shaped nanoparticles ($n_{1}=n_{2}=3.7$). Whenever η is large, the dimensionless velocity profile reaches the same value, which is limited to 1. Nanoparticle shape has a greater effect on velocity fields than temperature distribution. Figure 3 shows the effect of the shape factor on the temperature distribution of nanoparticles where $w_{1}=w_{2}=10\;gr,\;w_{f}=100\;gr$ = 0, σ = 0, Pr= 6.2, and γ =0. Figure 3 shows the effect of shape factor on temperature distributions of nanoparticles. In general, as the shape factor increases, temperature profiles decrease first and then increase, especially for disk nanoparticles. The image indicates that the shape factors of nanoparticles are not significantly different between these dimensionless temperature profiles. As η increases, the dimensionless temperature profile reaches a value limited to 0. Compared to temperature distribution, the shape factor of nanoparticles has a greater effect on the velocity field. Based on Figure 2 and Figure 3, increasing the levels of nanoparticle shape factors leads to increases in velocity and decreases in temperature. Consequently, the hydrodynamic boundary layer and thermal boundary layer become thinner. In these cases, the Prandtl number is fixed at 6.2, which does not account for variations.

When $w_{1}=w_{2}=10\;gr,\;w_{f}=100\;gr$ = 0, σ = 1, Pr= 6.2, γ = 0, the effects of shape factor of nanoparticles on velocity, and temperature distributions f′(η), g′(η), θ(η) of hybrid nanofluid under different values of shear–strain rate ratios γ=±3 are presented in Figure 4 and Figure 5. The behaviors of flow fields by changing shear–strain rate ratios γ were studied. f′(η),g′(η) increase with the increase in γ; θ(η) decreases with increase in γ. As shown in Figure 4, f′(η) reverse flows occur near walls at γ=−3, or g′(η) reverse flows occur near walls at γ=3, the flow is inward near the stagnation zone. As the shape factor increases, the velocities increase first and then decrease. Moreover, the velocity profile of disk nanoparticles is close to that of brick nanoparticles. Figure 5 shows influence of shape factor of nanoparticles on temperature distributions. As shape factor n increases, temperature profiles θ(η) decrease first and then increase; in particular, the temperature profile of disk nanoparticles is close to that of sphere nanoparticles because of the coefficient of permeability of porous medium σ=1 increasing. This dimensionless temperature profile under the influence of the nanoparticle shape factor shows no significant difference.

In nanofluids or hybrid nanofluids, the shape of nanoparticles affects both thermal characteristics and flow characteristics. A plot showing the influence of the shape factor n of nanoparticles on the skin friction coefficients and local Nusselt number can be seen in Figure 6. The skin friction coefficients of $Cf_{x},\;Cf_{y}$ and the local Nusselt number $Nu_{x}$ against γ in the range of −6≤γ≤6 for various shape factors $n_{1}=n_{2}=$3, 3.7, 4.8, 5.7, and 8.3 when $w_{1}=w_{2}$ = 10 gr, $w_{f}$ = 100 gr, σ = 0, and Pr=6.2 are shown in Figure 6. The values of $Cf_{x},\;Cf_{y}$ show a symmetric pattern where the line of symmetry lies at γ = 0 in the axisymmetric case. When γ = 0, σ = 0, and $w_{1}=w_{2}=0$, $Cf_{x}=f\prime \prime \left(0\right)$ = 1.311608, and $Cf_{y}=g\prime \prime \left(0\right)$ = 1.311608, also as shown in Table 5. As shown in Figure 6a, $Cf_{x}=Cf_{y}$ = 0; that is, the wall shear stress value is 0 when γ=±2.50895. When γ<−2.50895, $Cf_{x}$ decreases with increasing $w_{1},\;w_{2}$; when γ>−2.50895 increases, $Cf_{x}$ increases with increasing $w_{1},\;w_{2}$. When γ>2.50895 increases, $Cf_{y}$ decreases with increasing $w_{1},\;w_{2}$; when γ<2.50895, $Cf_{y}$ increases with increasing $w_{1},\;w_{2}$. Figure 6b shows the local Nusselt number $Nu_{x}$ against γ in the range of −6≤γ≤8 for various shape factors $n_{1}=n_{2}=$ 3, 3.7, 4.8, 5.7, and 8.3 when $w_{1}=w_{2}$ = 10 gr, $w_{f}$
= 100 gr, σ = 0, and Pr=6.2. $Nu_{x}$ increases as the value of the shape factor n increases. When γ<0, the values of $Nu_{x}$ increase sharply, with shape factor n increasing. When γ>0, the values of $Nu_{x}$ increase and slow down with an increase in shape factor n. In addition, for the axisymmetric case of γ = 0, when $w_{1}=w_{2}$ = 10 gr, $w_{f}$ = 100 gr, σ = 0, and Pr = 6.2, $Cf_{x}=Cf_{y}$ = 2.03443, 1.69439, 1.42415, 1.29710, and 1.63776, changing with the shape factor $n_{1}=n_{2}=$ 3, 3.7, 4.8, 5.7, and 8.3, as shown in Figure 7a. Among the five nanoparticle shapes, including spherical ($n_{1}=n_{2}=$ 3), brick-shaped ($n_{1}=n_{2}=$ 3.7), cylindrical ($n_{1}=n_{2}=$ 4.8), platelet-shaped ($n_{1}=n_{2}=$ 5.7), and disk-shaped ($n_{1}=n_{2}=$ 8.3), spherical and platelet-shaped nanoparticles have the highest and lowest friction coefficients, respectively. Skin friction $Cf_{x},Cf_{y}$ decreases as the shape factor n increases. Among the five nanoparticle shapes, including spherical ($n_{1}=n_{2}=$ 3), brick-shaped ($n_{1}=n_{2}=$ 3.7), cylindrical ($n_{1}=n_{2}=$ 4.8), platelet-shaped ($n_{1}=n_{2}=$ 5.7), and disk-shaped ($n_{1}=n_{2}=$ 8.3), the highest local Nusselt number is related to spherical nanoparticles, as shown in Figure 7b. $Nu_{x}$ increases with an increase in shape factor n, showing a close correlation of the five nanoparticle shapes.

When $w_{1}=w_{2}$ = 10 gr, $w_{f}$ = 100 gr, Pr=6.2
$n_{1}=n_{2}=3$ (spherical nanoparticles), the skin friction coefficients of $Cf_{x},\;Cf_{y}$ against γ in range of −6≤γ≤6 for various σ = 0, 1, 2, 5 are shown in Figure 8a. Additionally, for the axisymmetric case of γ = 0, when $w_{1}$ = $w_{2}$ = 10 gr, $w_{f}$ = 100 gr, $Cf_{x}=Cf_{y}$ = 2.03443, 2.27994, 2.50681, 3.10222 for various σ = 0, 1, 2, 5. Compared with $w_{1}=w_{2}$ = 10 gr, $w_{f}$ = 100 gr, σ = 0, it can be found that the wall shear stress values increase by 12.06%, 23.21%, and 52.48%, respectively. It is shown the percentage of skin friction coefficient enhanced by hybrid nanofluid relative to regular fluid. This shows that shear stress can be enhanced by increasing coefficient of permeability in porous media. As σ increases, the values of $Cf_{x}$ and $Cf_{y}$ increase. When $w_{1}=w_{2}$ and γ are given certain values, the greater the values of σ, the stronger the values of $Cf_{x}$ and $Cf_{y}$. When $w_{1}=w_{2}$ = 10 gr, $w_{f}$ = 100 gr, Pr=6.2 $n_{1}=n_{2}=3$ (spherical nanoparticles), local Nusselt number $Nu_{x}$ with γ in the range of −6≤γ≤8 for various σ = 0, 1, 2, 5 are shown in Figure 8b. As the coefficient of permeability in porous media increases, the values of $Nu_{x}$ decrease when γ<0. When γ>0, the values of $Nu_{x}$ are less affected by the values of σ.

When $w_{1}$ = 10 gr, $w_{f}$ = 100 gr, Pr=6.2 $n_{1}=n_{2}=3$ (spherical nanoparticles), skin friction coefficients of $Cf_{x},\;Cf_{y}$ with γ in the range of −6≤γ≤6 for various $w_{2}$ = 0, 10 gr, 30 gr, σ = 0, 1 are shown in Figure 9a. When $w_{1}$ = 10 gr, $w_{f}$ = 100 gr, σ = 0, $Cf_{x}=Cf_{y}$ = 1.40293, 2.03443, 2.42377 for various $w_{2}$ = 0, 10 gr, 30 gr. Compared with $w_{1}$ = 10 gr, $w_{2}$ = 0, $w_{f}$ = 100 gr, σ = 0, it can be found that the wall shear stress values increase by 45.01% and 72.76%, respectively. When $w_{1}$ = 10 gr, $w_{f}$ = 100 gr, σ = 1, $Cf_{x}=Cf_{y}$ = 1.70784, 2.27994, 2.68638 for various *w*2 = 0, 10 gr, 30 gr. Compared with $w_{1}$ = 10 gr, $w_{2}$ = 0, $w_{f}$ = 100 gr, σ = 0, it can be found that the wall shear stress values increase by 33.50% and 57.30%, respectively. This shows that shear stress can be augmented by increasing the mass of the second nanoparticle. When $w_{1}$ = 10 gr, $w_{f}$ = 100 gr, Pr=6.2 $n_{1}=n_{2}=3$ (sphere nanoparticle), local Nusselt number $Nu_{x}$ with γ in range of −6≤γ≤8 for various $w_{2}$ = 0, 10 gr, 30 gr, σ = 0, 1, 2 are shown in Figure 9b. As the mass of the second nanoparticle increases, $Nu_{x}$ increases more evenly. With an increase in the mass of the first and second nanoparticles of hybrid nanofluids, the relevant Nusselt number, and skin friction coefficient are enhanced. Since the mass of the nanoparticles increases, thermal conductivity is also augmented, affecting the rate of heat transfer.

## 4. Conclusions

The impact of the shape factor on mass-based hybrid nanofluid models for Homann stagnation-point flow in porous media was studied herein. The HAM-based Mathematica package BVPh 2.0 is suitable for determining the solution of coupled nonlinear ODEs with boundary conditions. Free online instructions for BVPh 2.0 are available (http://numericaltank.sjtu.edu.cn/BVPh.htm (accessed on 18 May 2013)). The analysis involved a discussion of the impact of the many physical parameters generated in the proposed model. The results show that the skin friction coefficients of *Cf_x,_* and *Cf_y_* increase with the mass of the first and second nanoparticles of hybrid nanofluids *w*_1_ and *w*_2_ and with the coefficient of permeability in porous media. For the axisymmetric case of γ = 0, when *w*_1_ = *w*_2_ = 10 gr and *w*_*f*_ = 100 gr, *Cf_x_* = *Cf_y_* = 2.03443, 2.27994, 2.50681, and 3.10222 for *σ* = 0, 1, 2, and 5. Compared with *w*_1_ = *w*_2_ = 10 gr, *w*_*f*_ = 100 gr, and *σ* = 0, it can be seen that the wall shear stress values increase by 12.06%, 23.21% and 52.48%, respectively. According to the calculation results, platelet-shaped nanoparticles in mass-based hybrid nanofluid models can achieve maximum heat transfer rates and minimum surface friction. As the mass of the first and second nanoparticles of hybrid nanofluid models increases, the local Nusselt number Nu_x_ increases. Nu_x_ decreases and obviously changes with an increase in the coefficient of permeability in the range of γ< 0; otherwise, Nu_x_ is less affected in the range of γ > 0.

## Figures and Tables

**Figure 1 nanomaterials-13-00984-f001:**
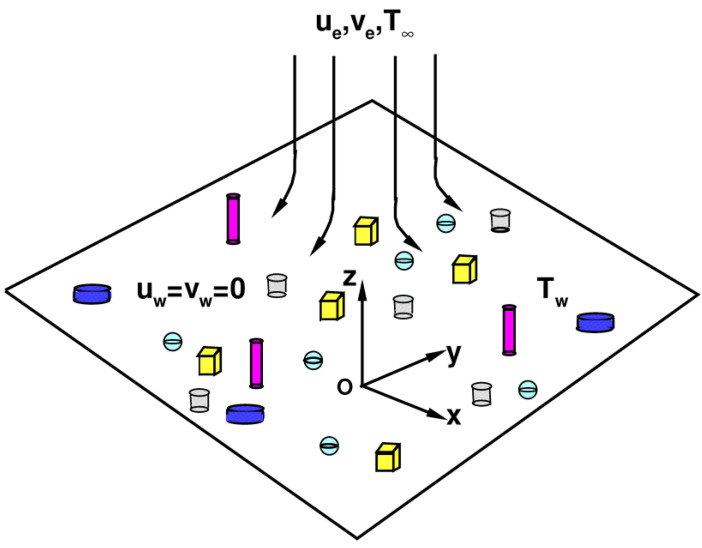
Physical model and coordinate systems.

**Figure 2 nanomaterials-13-00984-f002:**
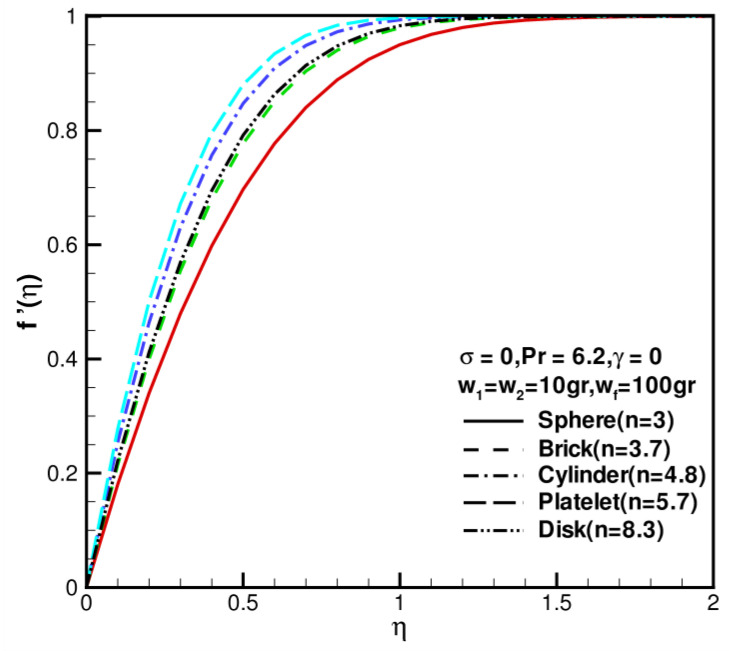
The dimensionless velocity component $f^{\prime }\left(\eta \right)$ against the shape factor *n* when $w_{1}$ = $w_{2}$ = 10 gr, $w_{f}$ = 100 gr, σ = 0, Pr = 6.2, and γ = 0.

**Figure 3 nanomaterials-13-00984-f003:**
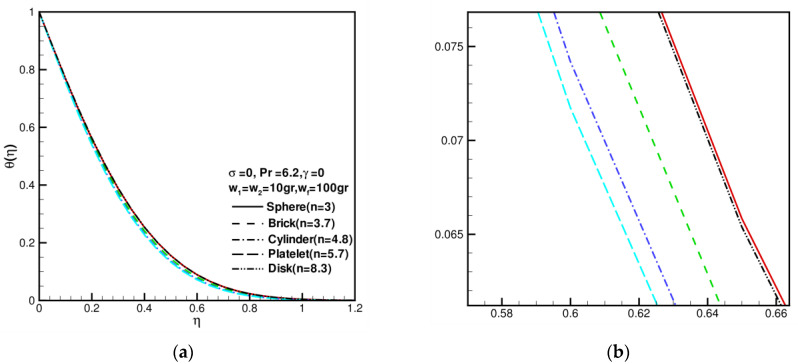
The dimensionless temperature distributions θ(η) against shape factor n when $w_{1}$ = $w_{2}$ = 10 gr, $w_{f}$ = 100 gr, σ = 0, Pr = 6.2, and γ = 0: (**a**) the full picture of θ(η); (**b**) the partial enlargement picture of θ(η).

**Figure 4 nanomaterials-13-00984-f004:**
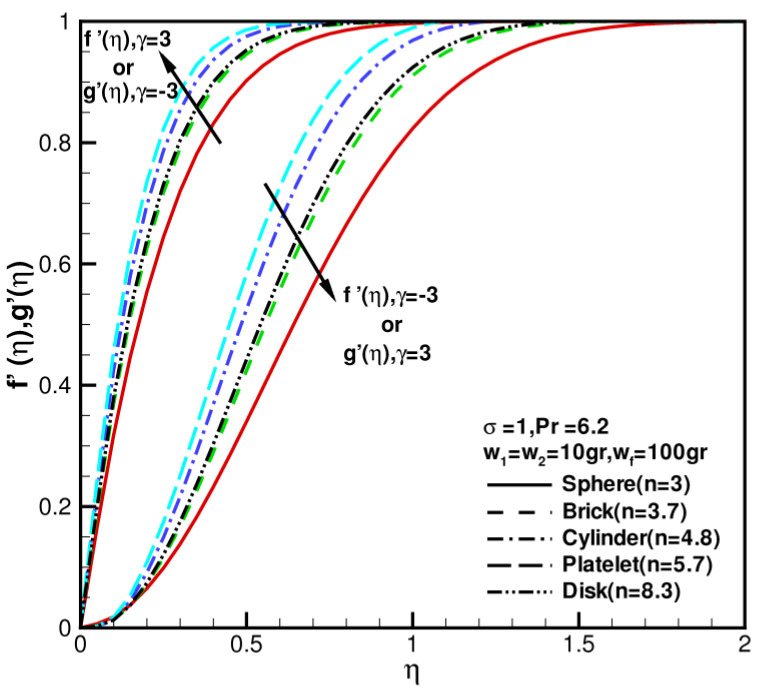
The dimensionless velocity components $f^{\prime }\left(\eta \right),\;g\prime \left(\eta \right)$ against shape factor n when $w_{1}$ = $w_{2}$ = 10 gr, $w_{f}$ = 100 gr, σ = 1, Pr = 6.2, and γ = −3, 3.

**Figure 5 nanomaterials-13-00984-f005:**
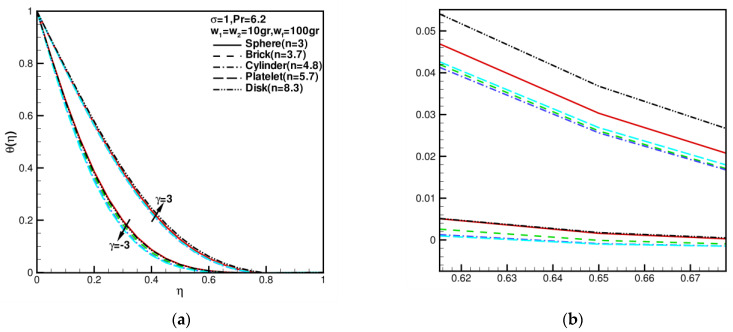
The variations in dimensionless fluid temperature component θ(η) against the shape factor n when $w_{1}$ = $w_{2}$ = 10 gr, $w_{f}$ = 100 gr, σ = 1, Pr = 6.2, and γ = −3, 3: (**a**) the full picture of θ(η); (**b**) the partial enlargement picture of θ(η); (**b**) the partial enlargement picture of θ(η).

**Figure 6 nanomaterials-13-00984-f006:**
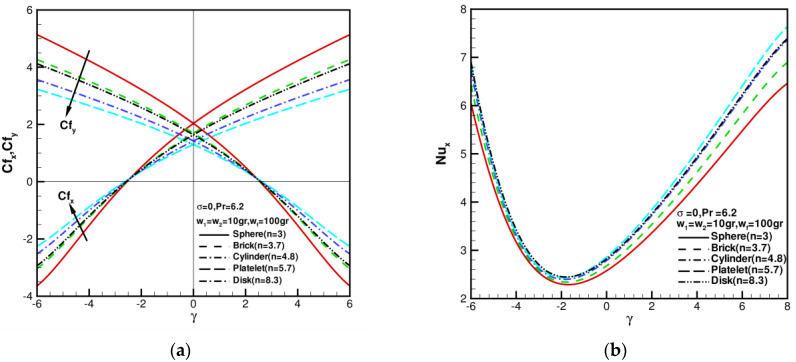
Skin friction coefficients $Cf_{x},\;Cf_{y}$ and local Nusselt number $Nu_{x}$ against γ, shape factor n when $w_{1}$ = $w_{2}$ = 10 gr, $w_{f}$ = 100 gr, σ = 0, and Pr = 6.2: (**a**) the distributions of $Cf_{x},\;Cf_{y}$; (**b**) the distributions of $Nu_{x}$.

**Figure 7 nanomaterials-13-00984-f007:**
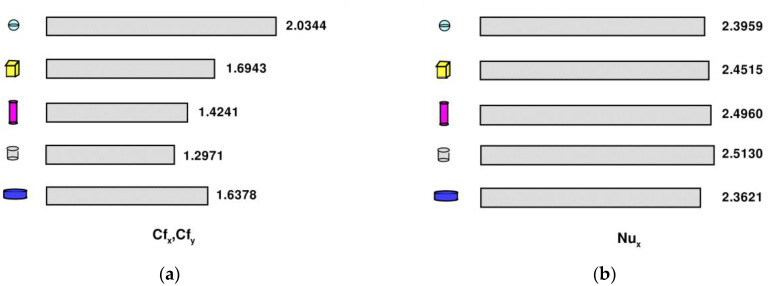
Skin friction coefficients $Cf_{x},\;Cf_{y}$ and local Nusselt number $Nu_{x}$ for various nanoparticle shapes when $w_{1}$ = $w_{2}$ = 10 gr, $w_{f}$ = 100 gr, σ = 0, and Pr = 6.2: (**a**) the distributions of $Cf_{x},\;Cf_{y}$; (**b**) the distributions of $Nu_{x}$.

**Figure 8 nanomaterials-13-00984-f008:**
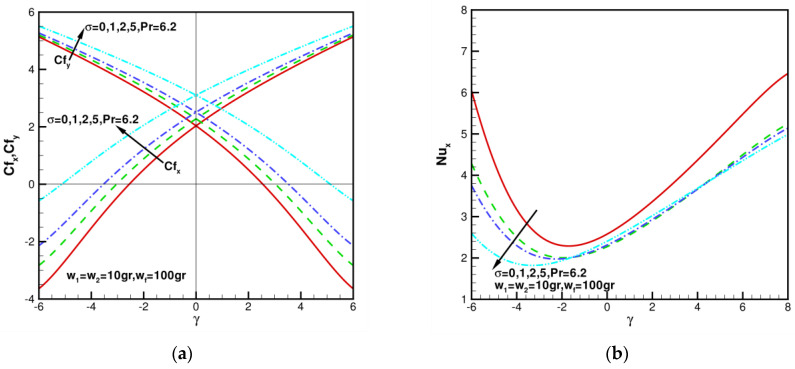
Skin friction coefficients $Cf_{x},\;Cf_{y}$ and local Nusselt number $Nu_{x}$ against γ when $w_{1}$ = $w_{2}$ = 10 gr, $w_{f}$ = 100 gr, σ = 0, 1, 2, 5, and Pr = 6.2: (**a**) the distribution of $Cf_{x},\;Cf_{y}$ with *n*1 = *n*2 = 3 (sphere); (**b**) the distribution of $Nu_{x}$ with $n_{1}=n_{2}=3$ (sphere).

**Figure 9 nanomaterials-13-00984-f009:**
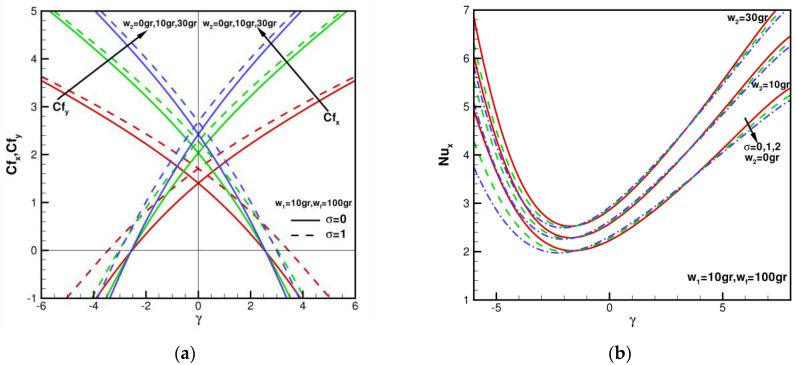
Skin friction coefficients $Cf_{x},\;Cf_{y}$ and local Nusselt number $Nu_{x}$ against γ when $w_{1}$ = 10 gr, $w_{f}$ = 100 gr, $w_{2}$ = 0, 10 gr, 30 gr, σ = 0, 1, 2, 5, and Pr = 6.2: (**a**) the distribution of $Cf_{x},\;Cf_{y}$ with $n_{1}=n_{2}=3$ (sphere); (**b**) the distribution of $Nu_{x}$ with $n_{1}=n_{2}=3$ (sphere).

**Table 1 nanomaterials-13-00984-t001:** Properties of H_2_O and nanoparticles [6,18,23,24].

Properties	$\rho \;({\mathrm{kg}/m}^{3})$	k (W/mK)	$c_{p}\;(J/\mathrm{kgK})$
H_2_O	997.1	0.613	4179
Cu	8933	401	385
Al_2_O_3_	3970	40	765

**Table 2 nanomaterials-13-00984-t002:** Mass-based hybrid nanofluid models for thermophysical properties [23,24,53].

Properties	Formulation
Heat capacitance	${\left(\rho c_{p}\right)}_{hnf}=\phi {\left(\rho c_{p}\right)}_{s}+\left(1-\phi \right){\left(\rho c_{p}\right)}_{f}$
Density	$\rho_{hnf}=\phi \rho_{s}+\left(1-\phi \right)\rho_{f}$
Dynamic viscosity	$\mu_{hnf}=\frac{\mu }{{\left(1-\phi \right)}^{2.5}}$ (Spherical)
	$\mu_{hnf}=\left(1+A\phi +B\phi^{2}\right)\mu_{f}$ (Non-spherical)
Thermal conductivity	$\frac{k_{hnf}}{k_{nf}}=\frac{k_{s2}+\left(n_{2}-1\right)k_{nf}-\left(n_{2}-1\right)\phi_{2}\left(k_{nf}-k_{s2}\right)}{k_{s2}+\left(n_{2}-1\right)k_{nf}+\phi_{2}\left(k_{nf}-k_{s2}\right)}$
	$\frac{k_{nf}}{k_{f}}=\frac{k_{s1}+\left(n_{1}-1\right)k_{f}-\left(n_{1}-1\right)\phi_{1}\left(k_{f}-k_{s1}\right)}{k_{s1}+\left(n_{1}-1\right)k_{f}+\phi_{1}\left(k_{f}-k_{s1}\right)}$

**Table 3 nanomaterials-13-00984-t003:** The mass-based models for selective hybrid nanofluids [23,24,53].

Properties	Mathematical Relations
Equivalent density	$\rho_{s}=\frac{\left(\rho_{1}\times w_{1}\right)+\left(\rho_{2}\times w_{2}\right)}{w_{1}+w_{2}}$
Specific heat equivalent of nanoparticles at constant pressure	${\left(c_{p}\right)}_{s}=\frac{\left({\left(c_{p}\right)}_{1}\times w_{1}\right)+\left({\left(c_{p}\right)}_{2}\times w_{2}\right)}{w_{1}+w_{2}}$
Solid volume fraction of first nanoparticle	$\phi_{1}=\frac{\frac{w_{1}}{\rho_{1}}}{\frac{w_{1}}{\rho_{1}}+\frac{w_{2}}{\rho_{2}}+\frac{w_{f}}{\rho_{f}}}$
Solid volume fraction of second nanoparticle	$\phi_{2}=\frac{\frac{w_{2}}{\rho_{2}}}{\frac{w_{1}}{\rho_{1}}+\frac{w_{2}}{\rho_{2}}+\frac{w_{f}}{\rho_{f}}}$
Equivalent volume fraction of nanoparticles	$\phi =\phi_{1}+\phi_{2}=\frac{\frac{w_{1}+w_{2}}{\rho_{s}}}{\frac{w_{1}+w_{2}}{\rho_{s}}+\frac{w_{f}}{\rho_{f}}}$

**Table 4 nanomaterials-13-00984-t004:** Sphericity, empirical shape factor, and A, B for non-spherical nanoparticles [18,23,24,25].

Nanoparticle Shape	Sphere	Brick	Cylinder	Platelet	Disk
n	3	3.7	4.8	5.7	8.3
ψ	1	0.81	0.62	0.52	0.36
A		1.9	13.5	37.1	14.6
B		471.4	904.4	612.6	123.3

**Table 5 nanomaterials-13-00984-t005:** Skin friction coefficients of $Cf_{x},\;Cf_{y}$ and local Nusselt number $Nu_{x}$ under values of γ, ϕ, and σ when Pr = 6.2.

γ	ϕ	σ	$Cf_{x}$ (Ref. [50])	HAM 20th	Relative Error(%)	$Cf_{y}$ (Ref. [50])	HAM 20th	Relative Error(%)	$Nu_{x}$ (Ref. [50])	HAM 20th	Relative Error(%)
0	0	0	1.311938	1.311608	0.0252	1.311938	1.311608	0.0252	1.806069	1.810147	0.2258
5			3.038940	3.036096	0.0935	−0.894909	−0.902242	0.8194	3.938146	3.998352	0.2257
−5			−0.894909	−0.902242	0.8194	3.038940	3.036096	0.0935	3.074275	3.084240	0.3241

## Data Availability

The manuscript includes all relevant data.

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
