# Peer review of "Shape-Factor Impact on a Mass-Based Hybrid Nanofluid Model for Homann Stagnation-Point Flow in Porous Media"

_nanomaterials, 2023, doi:10.3390/nano13060984_

Round 1
Reviewer 1 Report
Review report for the manuscript nanomaterials-2242735 entitled “Shape Factor Impact on Mass-based Hybrid Nanofluid Model for Homann Stagnation-point Flow in Porous Media”. Shape factor impact on mass-based hybrid nanofluid model for Homann stagnation-point flow in porous media is studied in this paper. The analysis involves discussions of impact of many physical parameters generated in the proposed model. Overall, the results are interesting and worthy to be published.
Please address the following comments:
1. English text needs to be reviewed.
2. The novelty of the present study needs to be declared in the abstract.
3. Add some quantitative data (i.e. percentage etc.) regarding improvement of results in the abstract.
4. In the introduction, the discussion regarding the nanoparticle shape factor is poor. I I suggest to improve this part and consider the following references as well.
(https://doi.org/10.1016/j.powtec.2018.08.086), (https://doi.org/10.1108/HFF-09-2018-0496)
5. The discussion regarding the figures 2 and 3 is poor. Please discuss technically about the changes.
6. Have you used commercial package or in-house code for conducting your numerical simulations? It must be clarified.
7. Which numerical method is used for numerical simulations? Only putting the governing equations is not enough.
8. The details of geometry and boundary conditions are required. So for, your simulation is not reproducible.
9. In the author contributions, I couldn’t understand who was responsible for the software, simulation, investigation and so on. Please clarify it.
Author Response
Q1.English text needs to be reviewed.
Response Q1: We apologize for the poor language of our manuscript. We worked on the manuscript for a long time and the repeated addition and removal of sentences and sections obviously led to poor readability. We have now worked on both language and readability and have also involved native English speakers for language editing services(MDPI editing services). We really hope that the flow and language level have been substantially improved. The corrected details are listed as highlighted in the revised manuscript.
Q2.The novelty of the present study needs to be declared in the abstract.
Response Q2: We deeply appreciate the reviewer’s suggestion. According to the reviewer’s comment, we have modified in the abstract. The corrected details are listed as highlighted in the revised manuscript.
Q3.Add some quantitative data (i.e. percentage etc.) regarding improvement of results in the abstract.
Response Q3: Our deepest gratitude goes to you for your careful work and thoughtful suggestions that have helped improve this paper substantially. According to the reviewer’s comment, we have added some quantitative data (i.e. percentage etc.) regarding of results in the abstract. The corrected details are listed as highlighted in the revised manuscript.
Q4.In the introduction, the discussion regarding the nanoparticle shape factor is poor. I suggest to improve this part and consider the following references as well.
(https://doi.org/10.1016/j.powtec.2018.08.086), (https://doi.org/10.1108/HFF-09-2018-0496)
Response Q4: We are grateful for the suggestion. To be more clear and in accordance with the reviewer concerns, we have added a brief description of some references as follows:
Q5.The discussion regarding the figures 2 and 3 is poor. Please discuss technically about the changes.
Response Q5: Thank you for your precious comments and advice. Those comments are all valuable and very helpful for revising and improving our paper. We have revised in the manuscript. The corrected details are listed as highlighted in the revised manuscript.
Q6.Have you used commercial package or in-house code for conducting your numerical simulations? It must be clarified.
Response Q6: Thank you very much for the reviewer’s suggestion. According to the reviewer’s comment, we have modified and the corrected details are listed as highlighted in the revised manuscript
Q7.Which numerical method is used for numerical simulations? Only putting the governing equations is not enough.
Response Q7: We deeply appreciate the reviewer’s suggestion. According to the reviewer’s comment, we have added a more detailed interpretation regarding the proper HAM results in the section 2. The corrected details are listed as highlighted in the revised manuscript.
Q8.The details of geometry and boundary conditions are required. So for, your simulation is not reproducible.
Response Q8: Thank you for your careful review. HAM-based Mathematica packages [48,49] are available for the solutions of coupled nonlinear ODEs (7) - (9) with boundary condition (10). Free online instructions for BVPh 2.0 are available online (http://numericaltank.sjtu.edu.cn/BVPh.htm).
Q9.In the author contributions, I couldn’t understand who was responsible for the software, simulation, investigation and so on. Please clarify it.
Response Q9: We deeply appreciate the reviewer’s suggestion. According to the reviewer’s comment, we have added a more detailed interpretation regarding the responsibilities in the author contributions. The corrected details are listed as highlighted in the revised manuscript.

Reviewer 2 Report
Shape Factor Impact on Mass-based Hybrid Nanofluid Model for Homann Stagnation-point Flow in Porous Media.
Introduction section is solid and concrete.
In the Materials section, a critical parameter for nanostructures' properties is size and morphology, how do the authors addressed this in the manuscript and simulations? I saw authors detail on Table 4, is this correct?
Any thoughts on filler fraction of reinforcing nanoparticles within the fluids? or is this subject out of the scope of the manuscript?
Even authors use some references from 2020-2021, i highly recommend to use more recent references if possible.
Author Response
Q1.Introduction section is solid and concrete.
Response Q1: We deeply appreciate the reviewer’s suggestion and encourage affirmation.
Q2.In the Materials section, a critical parameter for nanostructures' properties is size and morphology, how do the authors addressed this in the manuscript and simulations? I saw authors detail on Table 4, is this correct?
Response Q2: Our deepest gratitude goes to you for your careful work and thoughtful suggestions that have helped improve this paper substantially. The following section describes the size and morphology of nanoparticles in this manuscript. The data in Table 4 is referred to ref[23-25].
Q3.Any thoughts on filler fraction of reinforcing nanoparticles within the fluids? or is this subject out of the scope of the manuscript?
Response Q3: Thank you for your precious comments and advice. Those comments are all valuable and very helpful for revising and improving our paper. In both experimental and simulation studies from references, volume fraction, filler shape, and filler size were found to be the most significant factors that affect composite material thermal conductivity. We have added references related to the volume fraction of nano fillers as follows:
Q4.Even authors use some references from 2020-2021, I highly recommend to use more recent references if possible.
Response Q4: We are grateful for the suggestion. To be more clear and in accordance with the reviewer concerns, we have added a brief description of some references from 2022-2023 as follows:

Reviewer 3 Report
This article studied the shape factor impact on mass-based hybrid nanofluid model for Homann stagnation-point flow in porous media. The analysis involves discussions of impact of many physical parameters generated in the proposed model. Results have shown that skin friction coefficients of Cfx, Cfy increase with mass of first and second nanoparticles of hybrid nanofluid w1,w2 and the coefficient of permeability in porous media. As the mass of first and second nanoparticles of mass-based hybrid nanofluid model increases, local Nusselt number Nux increase. Values of Nux decrease and change obviously with the coefficient of permeability increasing in the range of γ<0; otherwise Nux are less affected in the range of γ>0. According to the calculation results, platelet nanoparticles of mass-based hybrid nanofluid model can achieve maximum heat transfer rate, and minimum surface friction.
First of all, the first affirmation from the introduction is wrong. A hybrid nanofluid means more, and that situation is a particular case.
English language is very poor.
Table 1 as well as Table 2 and related explanations have as reference 2 papers where these nanoparticles are not considered for the study. Nevertheless, the 2 articles are from the same group and the same errors are present, as for example the equations for estimating the hybrid nanofluids behaviour. Current state of the art clearly outlines for the last 3 years (at least) that for nanofluids and hybrid nanofluids the only correct approach is to use the experimental data for the thermophysical properties.
There is no nomenclature attached to this article and the notations are not explained in the manuscript.
The name of section 2 is not in accordance to its content.
Table 3: the first column is about nanoparticles and no correspondence with the formula from column 2 is noticed.
In section 2 is not clear what is the shape of these nanoparticles and how this is related to a certain application.
Results discussion is confusing.
Concluding, this paper cannot be published since contains many errors.
Author Response
Reviewer #1
This article studied the shape factor impact on mass-based hybrid nanofluid model for Homann stagnation-point flow in porous media. The analysis involves discussions of impact of many physical parameters generated in the proposed model. Results have shown that skin friction coefficients of Cfx, Cfy increase with mass of first and second nanoparticles of hybrid nanofluid w1,w2 and the coefficient of permeability in porous media. As the mass of first and second nanoparticles of mass-based hybrid nanofluid model increases, local Nusselt number Nux increase. Values of Nux decrease and change obviously with the coefficient of permeability increasing in the range of γ<0; otherwise Nux are less affected in the range of γ>0. According to the calculation results, platelet nanoparticles of mass-based hybrid nanofluid model can achieve maximum heat transfer rate, and minimum surface friction.
Q1.First of all, the first affirmation from the introduction is wrong. A hybrid nanofluid means more, and that situation is a particular case.
Response Q1: We deeply appreciate the reviewer’s suggestion. According to the reviewer’s comment, we have modified this expression in the introduction. We have now worked on both language and readability and have also involved native English speakers for language editing services(MDPI editing services). We really hope that the flow and language level have been substantially improved. The corrected details are listed as highlighted in the revised manuscript.
Q2.English language is very poor.
Response Q2: We apologize for the poor language of our manuscript. We worked on the manuscript for a long time and the repeated addition and removal of sentences and sections obviously led to poor readability. We have now worked on both language and readability and have also involved native English speakers for language editing services(MDPI editing services). We really hope that the flow and language level have been substantially improved. The corrected details are listed as highlighted in the revised manuscript.
Q3.Table 1 as well as Table 2 and related explanations have as reference 2 papers where these nanoparticles are not considered for the study. Nevertheless, the 2 articles are from the same group and the same errors are present, as for example the equations for estimating the hybrid nanofluids behaviour. Current state of the art clearly outlines for the last 3 years (at least) that for nanofluids and hybrid nanofluids the only correct approach is to use the experimental data for the thermophysical properties.
Response Q3: We deeply appreciate the reviewer’s suggestion. We have learned new knowledge, which can be applied to subsequent research. According to the reviewer’s comment, we have added the references for Table 1-4. About Table 1, we has cited four references [6,18,23,24] from different research groups. About Table 2 and 3, mass-based hybrid nanofluid model, we cited three references [23,24,50]. The corrected details are listed as highlighted in the revised manuscript.
Q4.There is no nomenclature attached to this article and the notations are not explained in the manuscript.
Response Q4: Thank you for underlining this deficiency. We has added the nomenclature attached to this manuscript. The corrected details are listed as highlighted in the revised manuscript.
Q5.The name of section 2 is not in accordance to its content.
Response Q5: Thank you for your careful review. We has modified as follow:
Q6.Table 3: the first column is about nanoparticles and no correspondence with the formula from column 2 is noticed.
Response Q6: Thank you for your careful review. We has modified as follow:
Q7.In section 2 is not clear what is the shape of these nanoparticles and how this is related to a certain application.
Response Q7: We are grateful for the suggestion. To be more clear and in accordance with the reviewer concerns, the corrected details of nanoparticle shapes are listed as highlighted in the revised manuscript as follows:
Q8.Results discussion is confusing.
Response Q8: We apologize for the poor language of our manuscript. We worked on the manuscript for a long time and the repeated addition and removal of sentences and sections obviously led to poor readability. We have now worked on both language and readability and have also involved native English speakers for language editing services(MDPI editing services). We really hope that the flow and language level have been substantially improved. Based on the three reviewers’ comments, we has modified in major revision. The corrected details are listed as highlighted in the revised manuscript
Q9.Concluding, this paper cannot be published since contains many errors.
Response Q9: We apologize for the poor language of our manuscript. We worked on the manuscript for a long time and the repeated addition and removal of sentences and sections obviously led to poor readability. We have now worked on both language and readability and have also involved native English speakers for language editing services(MDPI editing services). We really hope that the flow and language level have been substantially improved. Based on the three reviewers’ comments, we has modified in major revision. The corrected details are listed as highlighted in the revised manuscript.

Round 2
Reviewer 1 Report
accept
Reviewer 3 Report
I cannot see much improvement in the manuscript, despite its fine refinement in some places. No major changes are noticed in accordance to previous comments. So, I am keeping my decision to reject this article since contains errors.